# Characterizations and Antibacterial Efficacy of Chitosan Oligomers Synthesized by Microwave-Assisted Hydrogen Peroxide Oxidative Depolymerization Method for Infectious Wound Applications

**DOI:** 10.3390/ma14164475

**Published:** 2021-08-10

**Authors:** Vinh Khanh Doan, Khanh Loan Ly, Nam Minh-Phuong Tran, Trinh Phuong-Thi Ho, Minh Hieu Ho, Nhi Thao-Ngoc Dang, Cheng-Chung Chang, Hoai Thi-Thu Nguyen, Phuong Thu Ha, Quyen Ngoc Tran, Lam Dai Tran, Toi Van Vo, Thi Hiep Nguyen

**Affiliations:** 1Department of Tissue Engineering and Regenerative Medicine, School of Biomedical Engineering, International University, Ho Chi Minh City 700000, Vietnam; khanhvinh221296@gmail.com (V.K.D.); llkhanh2303@gmail.com (K.L.L.); tmpnambme@gmail.com (N.M.-P.T.); phuongtrinh79@gmail.com (T.P.-T.H.); hohieuminh96@gmail.com (M.H.H.); dntnhi@hcmiu.edu.vn (N.T.-N.D.); vvtoi@hcmiu.edu.vn (T.V.V.); 2Vietnam National University, Ho Chi Minh City 700000, Vietnam; ntthoai@hcmiu.edu.vn; 3Graduate Institute of Biomedical Engineering, National Chung Hsing University, Taichung City 40227, Taiwan; ccchang555@dragon.nchu.edu.tw; 4School of Biotechnology, International University, Ho Chi Minh City 700000, Vietnam; 5Institute of Materials Science, Vietnam Academy of Science and Technology, Ha Noi 100000, Vietnam; thuhp@ims.vast.ac.vn; 6Institute of Applied Materials Science, Vietnam Academy Science and Technology, Ho Chi Minh City 700000, Vietnam; tnquyen@iams.vast.vn; 7Graduate University of Science and Technology Viet Nam, Vietnam Academy of Science and Technology, Ho Chi Minh City 700000, Vietnam; 8Institute for Tropical Technology, Vietnam Academy of Science and Technology, 18 Hoang Quoc Viet, Cau Giay, Ha Noi 100000, Vietnam; tdlam@itt.vast.vn; 9Graduate University of Science and Technology, Vietnam Academy of Science and Technology, 18 Hoang Quoc Viet, Cau Giay, Ha Noi 100000, Vietnam

**Keywords:** chitosan oligomers (OCS), poly(ɛ-caprolactone) (PCL), microwave irradiation, electrospinning, tablet-compressing, multi-spraying coating, wound dressing

## Abstract

The use of naturally occurring materials with antibacterial properties has gained a great interest in infected wound management. Despite being an abundant resource in Vietnam, chitosan and its derivatives have not yet been intensively explored for their potential in such application. Here, we utilized a local chitosan source to synthesize chitosan oligomers (OCS) using hydrogen peroxide (H_2_O_2_) oxidation under the microwave irradiation method. The effects of H_2_O_2_ concentration on the physicochemical properties of OCS were investigated through molecular weight, degree of deacetylation, and heavy metal contamination for optimization of OCS formulation. Then, the antibacterial inhibition was examined; the minimum inhibitory concentration and minimum bactericidal concentration (MIC and MBC) of OCS-based materials were determined against common skin-inhabitant pathogens. The results show that the local Vietnamese chitosan and its derivative OCS possessed high-yield purification while the molecular weight of OCS was inversely proportional and proportional to the concentration of H_2_O_2_, respectively. Further, the MIC and MBC of OCS ranged from 3.75 to less than 15 mg/mL and 7.5–15 mg/mL, respectively. Thus, OCS-based materials induce excellent antimicrobial properties and can be attractive for wound dressings and require further investigation.

## 1. Introduction

Chitosan is a derivative of chitin, the second abundant biopolymer found in shrimp and crab shells. Chitosan mainly comprises of linear N-acetyl glucosamine and β-1,4-linked D-glucosamine units. Chitosan has been broadly exploited in a variety of fields, including agriculture, food industry, water treatment, biotechnology, pharmaceutics, and many others, thanks to its pronounced biocompatibility, biodegradability, low cost, and non-toxicity [1,2,3,4,5]. Chitosan is highly viscous in aqueous solution due to its high molecular weight and is a prominent pH-responsive biopolymer, which is soluble in mild acidic solution with pH below 6.3 while becomes insoluble with gel-forming ability at physiological pH (~7–7.4) [6,7,8]. These properties, however, may limit chitosan applicability in medicine and biomedical studies that involve physiological conditions. Therefore, chitosan oligomers (OCS), derived from chitin or chitosan, with high water solubility thanks to shorter chain and free amino (-NH_2_) groups in D-glucosamine units, have emerged as a promising alternative [9,10]. Furthermore, OCS also exhibit bioactive properties such as antimicrobial, anti-inflammatory, antifungal, antitumor, and so on, making them highly desired for biomedical applications such as wound healing scaffolds or regenerative medicine.

In terms of the fabrication method, a variety of chemical, enzymatic, and physical processes were proposed [11,12,13]. The enzymatic approach to degrade chitosan into OCS is simple and environmentally friendly, yet its production cost can be exorbitant [14,15]. Meanwhile, the physical approaches using ultrasonic, microwave, and gamma rays can be employed for the synthesis of high purified OCS [16,17,18,19]. However, it remains difficult to scale up the production of OCS using physical methods due to the lack of corresponding facilities [20]. On the other hand, the chemical approach, such as oxidative degradation of chitosan using hydrogen peroxide (H_2_O_2_), which can be used for large-scale production of OCS at a reasonable cost, has emerged as a promising alternative and received a great research interest [21,22,23]. Nevertheless, the relative molecular weight of OCS produced by this method is widely distributed, thus, requires intensive post-treatment to separate and purify the products [11]. Qin et al. enhanced the oxidative degradation of chitosan for the production of OCS using conventional heating-combined H_2_O_2_ treatment [24]. Najafabadi et al. proposed a UV irradiation-H_2_O_2_ system to enable a faster and more efficient OCS production [25]. Besides, to improve the selectivity, reaction rate, and production efficiency, several studies utilized microwave irradiation as an effective solution for enhanced oxidative degradation of chitosan chains [23,26]. In the current study, the use of microwave-assisted H_2_O_2_ treatment for the production of OCS was employed for convenience as well as time- and cost-saving benefits.

Vietnam was one of the largest shrimp exporting countries in the world, with a network worth billions of dollars per year [27]. However, the crustacean processing may lead to the tremendous gross weight of shell waste, causing a large environmental burden. For both economic and environmental benefits, the use of local resources or bio-waste of the crustacean processing for the production of OCS emerged as a promising and sustainable solution. Several studies in Vietnam attempted to synthesize OCS using different approaches such as H_2_O_2_ degradation and gamma irradiation combined with H_2_O_2_ treatment for mainly agricultural applications [18,19,21,28]. Nevertheless, studies utilizing the local chitin source to produce OCS for infectious wound management purposes are lacking.

In this study, we aim to investigate the antibacterial efficacy of OCS derived from a local Mekong Delta, Vietnam source using microwave-assisted H_2_O_2_ treatment as a promising wound dressing material. First, the concentration of H_2_O_2_ was varied to determine its effects on the physicochemical properties of the synthesized OCS. Then, we examined the minimum inhibitory concentration and minimum bactericidal concentration (MIC and MBC) of OCS using dilution test and investigated the inhibitory effects of OCS-based tablets using agar diffusion test to determine the optimal OCS formulation. Recently, the surface modification or coating of antibacterial agents onto wound dressing has become a key approach for the fabrication of bacterial-preventive materials [28,29]. To demonstrate the applicability of the synthesized OCS for infectious wound management, the optimal OCS formulation was coated onto electrospun poly(ɛ-caprolactone) (EsPCL) membrane using the multi-immersion technique [30] and investigated the antibacterial inhibition of the OCS-coated EsPCL (EsPCLOCS) membranes. The findings suggest that the combination of this versatile synthesis and local supplied chitosan produces high-purified OCS with excellent antibacterial activities is promising and beneficial in terms of economic efficiency. Further, the EsPCLOCS membranes exhibit good antibacterial effects and require further investigations as potential wound dressing materials.

## 2. Materials and Methods

### 2.1. Material

Chitosan (low viscosity 150–250 cP, molecular weight of 311 kDa, and degree of deacetylation (DD) of 90.8%) from shrimp shells were obtained from Dao Nguyen Co., Ho Chi Minh City, Vietnam. Poly(ɛ-caprolactone) (PCL, Mn 80,000) was purchased from Sigma–Aldrich Co., St Louis, MO, USA. Acetic acid (CH3COOH, 99.5%), acetone (CH3COCH3, 99.5%), hydrogen peroxide (H2O2, 30% (*w*/*w*)) reagent and absolute ethanol (EtOH) were purchased from Xilong Chemical Co., Ltd. (Shantou, China). *Staphylococcus aureus* (*S. aureus*) ATCC 25213, *Pseudomonas aeruginosa* (*P. aeruginosa*) ATCC 9027, *Streptococcus iniae* (*S. iniae*)*, Candida albicans* (*C. albicans*)*, Trichosporon insectorum* (*T. insectorum*) were provided by Marine Laboratory, International University-HCM Vietnam National University, Ho Chi Minh City, Vietnam. Mueller Hinton broth (M391-500G; MHB) was purchased from Hi-Media (Maharastra, India). Other chemicals can be purchased at major suppliers.

### 2.2. Methodology

#### 2.2.1. Preparation of Chitosan Oligomers (OCS)

The OCS preparation method was reported elsewhere with some modifications [23]. Chitosan powder was immersed into different concentrations (5%, 10%, and 15% *v*/*v*) H2O2 solutions at 30 °C in 10 min. The mixture was microwave-irradiated at 400 W for 3 min, and the solution was then cooled down to room temperature. The solution was introduced into EtOH at a volume ratio of 1:3 and using Sigma 3-30KS at the speed of 10,000 rpm× *g* at 4 °C for 15 min to collect the precipitant, which was then lyophilized with LABCONCO 7,752,020 series to obtain OCS powders. The OCS powders were compressed into tablets with a mass of 120 mg and 13 mm in diameter by the Hydraulic Single tablet punching machine (Shanghai Pharmaceutical Machinery Co. Ltd., Shanghai, China) for further investigations.

#### 2.2.2. Characterization of Chitosan Oligomers

##### Gel Permeation Chromatography (GPC)

The weight average (M_w_), number average (M_n_) of molecular weight, and polydispersity index (PDI) of OCS were measured by Gel Permeation Chromatography (GPC) (Shimadzu/LC-10ADvp, Kyoto, Japan) with refractive index detector RID-10A. The system was carried out in the water as a mobile phase. All the samples were dissolved at 1.5 mg/mL in 0.3 M acetic acid and 0.2 M sodium acetate and filtered before GPC measurement with a flow rate of 0.8 mL/min at 40 °C with a sample volume of 20 μL. The Pullulan standards with a molecular weight range from 1.42 to 1220 kDa were used for calibrating OHpak SB-804 HQ columns (dimension 8 mm × 300 mm).

The depolymerization efficiency (*DE*) is calculated based on the following formula:$$DE=\frac{Initial\;molecular\;weight}{Final\;molecular\;weight}\times 100$$

##### Nuclear Magnetic Resonance (NMR) Spectrometer

The ^1^H-NMR spectrum of COS was measured by using liquid-state ^1^H-NMR (400 MHz, δ in ppm; Bruker Avance-400 MHz FT-NMR (Bruker Corp, Billerica, MA, USA). The chitosan oligomers samples were dissolved in DMSO/DCl and filtered prior to NMR measurement. The DD of OCS was then calculated based on the following equation [31]:$$\mathrm{DD}\left(\%\right)=\left(1-\frac{\frac{1}{3}A_{2}}{\frac{1}{6}A_{1}}\right)\times 100$$
where A_1_ are the protons integral values of positions C_2_–C_6_ on the sugar ring, which was the average area measured in the range δ 3–6 ppm, and A_2_ are the protons integral values of the three N-acetyl protons of N-Acetyl glucosamine at around δ 2 ppm.

##### Inductively Coupled Plasma Mass Spectroscopy (ICP-MS)

Chitosan and the OCS solution were evaluated for the contamination of lead (Pb), arsenic (As), and mercury (Hg), which are common heavy metals detected in shrimp shells-extracted products, by using ICP-MS (NexION2000, Perkin Elmer, MA, USA). The minimum limit of detection of all the contaminants is 0.02 ppm.

#### 2.2.3. Preparation and Morphological Characterization of EsPCLOCS Membrane

##### Preparation of EsPCLOCS Membrane

The EsPCL membrane was fabricated as previously reported [32]. Briefly, PCL was dissolved in acetone/acetic acid solution with *v/v* ratio of 7:3 in 24 h to create 22% (*w*/*v*) PCL solution. Then the solution was filled into a syringe pump and electrospun with a tip-to-collector distance of 10 cm and voltage of 15 kV. For fabricating EsPCL coated with OCS (EsPCLOCS) membrane, EsPCL membranes (50 cm × 30 mm) were plasma-treated with Harrick Plasma Cleaner PDC- 32G-2 (GaLa Instrumente, Bad Schwalbach, Hawai, Germany) for 3 min at 30 W and 13.56 MHz. To prepare the EsPCLOCS membrane, a 3% *w*/*v* OCS solution was obtained by dissolving OCS15% powder in deionized water, then was sprayed perpendicularly on the surface of the EsPCL membranes. Then, the samples were incubated at 37 °C for 30 min for the adsorption of OCS onto the fibers. Three different EsPCLOCS samples were fabricated, labeled as C1, C3, and C6, corresponding to the one, three, and six coatings. All the samples were sterilized by UV irradiation for 45 min before further antibacterial tests.

##### Morphological Characterization

The EsPCL and EsPCLOCS membranes were observed using scanning electron microscopy (SEM) (JSM-IT100, JEOL, Tokyo, Japan) with gold sputter-coating (JEOL Smart Coater, Tokyo, Japan) at 10 kV to evaluate the morphology of the membrane and the OCS layer covered on EsPCL fibers after each coating time.

#### 2.2.4. Antibacterial Assays

In the bacterial experiments, agar disc diffusion MIC/MBC methods were applied to determine the antibacterial effects of OCS on five strains of skin-habitant microorganisms, including *S. aureus, P. aeruginosa,* and *S. iniae* bacteria and *C. albicans* and *T. insectorum* fungi. For EsPCLOCS membranes, the antibacterial properties evaluation was performed with *S. aureus* and *P. aeruginosa.* Prior to the experiments, a colony of each strain was collected from an agar plate, transferred to a 5 mL MHB tube, and cultured at 37 °C for 24 h. Then bacterial suspension of each strain at an optical density at 620 nm OD_620_ = 0.08–0.1 (equal to 0.5 McFarland standards, approximately 1–2 × 10^8^ CFU/mL) was obtained by dilution.

##### Agar Disk Diffusion

The inhibitory effect of each sample against a specific bacterial strain was examined separately on the Mueller–Hinton agar (MHA) plate. Briefly, 150 µL of the prepared bacterial suspension was added into and spread on the MHA surface. Then, the OCS tablets and EsPCLOCS membranes with a diameter of 13 mm and 8 mm, respectively, were placed on the MHA plate and incubated for 24 h. The bacterial growth inhibition zone around the samples was determined.

##### Minimum Inhibitory Concentration (MIC) and Minimum Bactericidal Concentration (MBC)

Determination of MICs of the OCS15% was carried out on a 96-well plate, with up to 10 different dilution concentrations were tested per row. The OCS15% powder was dissolved in MHB for 24 h to obtain 2X solution (60 mg/mL). One hundred microliters of MHB were first dispensed into all wells. The 2X antibacterial solution was added to the first well and mixed well with MHB. Then 100 µL of the mixture in the first well was transferred to the corresponding well, and the process was repeated to obtain the 100 µL of MHB containing two-fold dilutions of OCS15% in the first ten wells. Finally, 100 µL of prepared bacterial suspension was added in columns 1st to 11th. After that, the microplate was incubated at 37 °C for 24 h. The absorbance was obtained using a microplate reader at a wavelength of 620 nm. MIC results were determined as the lowest concentration where no growth is observed. To determine MBC, 10 µL of the mixture from the first to the tenth well was inoculated in a circle on an MHA plate following a clockwise direction, while 10 µL of the 11th well was plated in the middle. The MHA plate was incubated right side up at 37 °C for 24 h. MBC will correspond to the lowest concentration with no observation of bacterial growth on the MHA [33].

#### 2.2.5. Statistical Analysis

All experiments were conducted in triplicate unless specified otherwise. Statistical analysis was performed by using Sigma Plot V.12.0 version (SSI, Chicago, IL, USA). The differences between samples were analyzed by one-way analysis of variance (ANOVA) followed by Tukey Kramer post hoc test. The data were expressed as the mean ± standard deviation, and *p* ˂ 0.05 was considered to be statistically significant.

## 3. Results

### 3.1. Characterizations of Chitosan Oligomers (OCS)

The need for highly purified chitosan is highly desired for biomedical applications. According to ICP-MS results shown in Table 1, only the sample of chitosan found the appearance of toxic metals; meanwhile, their traces were not detected in OCS. Chitosan contained 0.052 ppm of lead and 0.05 ppm of arsenic in the sample. Meanwhile, the toxic concentration of lead, arsenic, and mercury are reported to be 0.5, 0.15, and 1.5 ppm, respectively, according to United States Pharmacopeia for Food and Drugs. Thus, based on the ICP-MS results, there was an insignificant level of two of those heavy metals (lead and arsenic) detected in chitosan from the local Vietnamese brand.

OCS synthesized using hydrogen peroxide oxidation under microwave irradiation was examined by evaluating its weight average (M_w_), number average (M_n_), and the polydispersity index showed in Table 2. The method showed high effectiveness in terms of depolymerization. The molecular weight of raw chitosan was measured as 310 kDa and dropped significantly from 4500 to 12,600%. The molecular weight of OCS decreases gradually as the hydrogen peroxide concentration raised. The highest molecular weight among the three samples belongs to OCS5%, with an average of 7 kDa. Meanwhile, the average M_w_ of OCS10% is about 5 kDa, which is approximately double that of OCS15%. Appendix A shows that the distribution of M_w_ of OCS5% and OCS10% is overlapped with a considerably broad curve. In particular, OCS10% has the range of M_w_ from 178 Da to 20 kDa with PDI of 2.5, while OCS5% has the range of M_w_ distributing under 25 kDa. Meanwhile, OCS15% reveals a narrow distribution curve with the M_w_ under 14 kDa and PDI of 2.16. Figure 1 shows the ^1^H-NMR spectra of OCS15%. The integral values are determined by H molecules from the C2–C6 in the sugar ring of chitosan monomers (H_2–6_) from δ 2.6 to 4.2 ppm and N-acetyl methyl group with δ around 2 ppm. The DD was calculated based on the integral value in the H-NMR figures. The data showed that all the samples had DD over 90%. Furthermore, the method tended to slightly increase the DD of the samples compared to the original chitosan. However, there were no significant differences in the DD of three samples, in which the highest DD belonged to OCS15% (95.71%), whilst OCS5% and OCS10% had the lower DD of around 92%.

### 3.2. Antibacterial Properties

Table 3 shows the MIC and MBC concentrations of OCS15% against five different pathogens. In general, the MIC and MBC of OCS were closely comparable regardless of pathogen types. The MIC of all the bacteria and fungi were not precisely determined except for *P. aeruginosa* at 3.75 mg/mL. The MIC concentrations of OCS15% can be found in a range of 3.75 and 7.5 mg/mL with *T. insectorum* and 7.5 to 15 mg/mL in the case of *S. aureus*, *P. aeruginosa*, and *C. albicans*. OCS was proved to have bactericidal and fungicidal properties rather than just bacteriostatic or fungistatic through MBC. *S. iniae* and *T. insectorum* was killed when the concentration of OCS reached 7.5 mg/mL, while *S. aureus*, *P. aeruginosa,* and *C. albicans* required 15 mg/mL of OCS.

The antimicrobial effects of OCS tablets on various bacteria and fungi are shown in Figure 2. All the OCS specimens exhibited inhibition against all pathogens. It seemed that the Gram-negative pathogen (*P. aeruginosa*) responded less sensitive, whilst fungi, especially *T. insectorum*, were vulnerably exposed by OCS. Moreover, in terms of Gram-positive pathogens, similar zones of inhibition were seen in both *S. aureus* and *S. iniae*. Among those samples, OCS10% had the highest inhibitory effect on pathogens with the range of inhibitory zone from approximately 30 mm against *P. aeruginosa* to over 40 mm against *C. albicans*, whereas OCS15% exhibited less efficiency against those microorganisms.

The absorption of OCS on the EsPCL membrane before and after several coatings was evaluated by SEM micrographs. Figure 3A shows the differences in morphology surface of EsPCL and EsPCLOCS membranes. In general, the fiber surfaces demonstrated that the amount of adsorbed OCS onto treated EsPCL membrane increased proportionally to increased spraying time (Figure 3A(1)–(4)). In comparison to fiber diameters of EsPCL in Figure 3A(1), PCL fiber diameters enlarged gradually associated with the coating times. Moreover, the surface of EsPCL membranes was smoother than that of the EsPCLOCS membranes. In particular, OCS agglutinated around PCL fibers (Figure 3A(2)), then covered the pores on the surface membrane (Figure 3A(3)), and finally provided a complete coating layer over the whole membrane (Figure 3A(4)).

The antibacterial property of EsPCLOCS membranes with different OCS coating times was determined using the agar disc diffusion method. Figure 3B displays the zone of inhibition of EsPCLOCS membranes against *P. aeruginosa* and *S. aureus*. The results show that both pathogens were inhibited in a coating-layer-dependent manner except sample C1, which showed no inhibitory effect against *P. aeruginosa*, and the Gram-negative strain was found to be more susceptible to the coated-membrane than the positive. In particular, it is clear that the increment of OCS coating layers onto the surface of EsPCL membranes led to a significantly higher inhibitory effect against the pathogen, in which C3 and C6 have the inhibition zone diameter of 9 ± 0.3 mm and 10.2 ± 0.4 mm against *S. aureus*, respectively.

## 4. Discussion

In this study, we prepared OCS with different molecular weights by adjusting the H_2_O_2_ concentration. H_2_O_2_ oxidation under the support of microwave irradiation is a simple, inexpensive, and rapid method to enhance water solubility of chitosan by cutting the polymer chains of chitosan, to synthesize OCS. Based on the GPC results, the proposed method broke the chitosan polymer chains down to a molecular weight of around a few hundred Da to a few thousand Da, making it appropriate for antibacterial applications [34]. The main principle of oxidative degradation is based on the unstable of H_2_O_2_ molecules. The H_2_O_2_ could be easily decomposed under microwave and heat into hydroperoxyl anion ($HOO^{-})$, which later will interact with hydrogen peroxide molecules in order to form hydroxyl radical (*HO*∙) and superoxide anion ($O_{2}^{-})$. These two oxidants are more powerful to cut the polymer chain through the following reactions [26,35]:$$H_{2}O_{2}\to H^{+}+HOO^{-}$$
$$HOO^{-}+H_{2}O_{2}\to O_{2}^{-}+HO+H_{2}O$$
$${\left(GlcN\right)}_{m}-{\left(GlcN\right)}_{n}+HO\;\to {\left(GlcN\right)}_{m}-{\left(GlcN\right)}_{n}+H_{2}O$$
$${\left(GlcN\right)}_{m}-{\left(GlcN\right)}_{n}+H_{2}O\to \;{\left(GlcN\right)}_{m}+{\left(GlcN\right)}_{n}$$
where, ${\left(GlcN\right)}_{m}\;\mathrm{and}\;{\left(GlcN\right)}_{n}$ are chitosan chain that have m and *n* glucosamine molecules, respectively.

Notably, the degradation of chitosan into OCS is dependent on the concentration of H_2_O_2_. It can be seen that the 5% concentration of H_2_O_2_ inadequately degraded the polymer, mainly because of the shortage of H_2_O_2_ to interact with chitosan. Meanwhile, the DD is unlikely to be affected by this microwave irradiation in combination with the H_2_O_2_ treatment method, which is in contrast with other conventional methods [36]. The DD was not affected in this study can be due to the rapid reaction time accelerated by microwave; the H_2_O_2_ molecules and microwave power only interact with chitosan chains to degrade the polymer chains rather than continuously react with the -NH_2_ groups of chitosan.

Furthermore, microwave irradiation was used in this study in combination with H_2_O_2_ treatment to improve the reaction rate and production efficiency of OCS as well as to narrow the distribution range of OCS molecular weight. The average M_w_ of OCS reduced significantly from 7 kDa to 2 kDa as H_2_O_2_ concentration increased from 5% to 15%. The trend is similar to previous studies employing microwave-assisted H_2_O_2_ treatment under different experimental conditions (i.e., H_2_O_2_ concentration, microwave reaction time, microwave power, and so on) [20,23,37]. In this study, we used a relatively high concentration of H_2_O_2_ that significantly reduced the reaction time to only 3 min as compared to the previous study that used a low concentration of H_2_O_2_ and required a longer reaction time (more than 10 min) to obtain OCS of less than 2 kDa [20]. Further, the DD and the M_w_ of OCS5% obtained in this study was much higher and lower, respectively, compared to these values reported in Zhang et al. that used shorter microwave time (75 s) although higher microwave power (650 W) and lower H_2_O_2_ concentration (2%) [37].

In terms of antimicrobial effects, the pathogens used for in vitro antibacterial assays, including *S. aureus*, *S. iniae*, *P. aeruginosa, C. albicans*, and *T. insectorum,* are five common skin-habitant microorganisms [38,39,40,41]. In the moist environment of the agar disc, the OCS started to dissolve and diffuse after placing on the surface of the agar due to its high water-solubility and induced inhibitory effects against tested pathogens. Notably, OCS of different molecular weights induced different inhibitory effects. OCS15% with M_w_ of 2–3 kDa tends to have a relatively low activity to all pathogens except *C. albicans,* although it has the best water solubility. On the other hand, OCS10%, which has M_w_ around 5 kDa, provides the strongest influence in all pathogens. With M_w_ of 7 kDa, OCS5% shows limited activity on all types of microorganisms except *T. insectorum*. The results agreed with a previous study in which different pathogens react differently with different M_w_ of OCS [42]. Further, we noticed that the effect of OCS on pathogens might be different depending on their form. In particular, MIC results of OCS15% show the higher impacts on *P. aeruginosa* than Gram-positive *S. aureus*, only 3.75 mg/mL compared to about 7.5 mg/mL. However, in the agar diffusion test, there were no notable differences between the inhibitory zone of these strains. Similarly, *C. albicans,* which was strongly preventive to OCS solution with MIC of about 7.5 mg/mL, tended to be the most vulnerable in the agar test with the largest zone of inhibition. While chitosan is believed to be only bacteriostatic, its derivative, OCS, is not only exhibiting bacteriostatic but also revealing as a bactericide and fungicide. This characteristic was rarely reported due to lacking concern. Lee and colleagues previously claimed that 10 kDa OCS of 1 to 10 mg/mL could have stronger bactericidal effects on *V. vulnificus,* a life-threatening wound infection pathogen, than 1 kDa OCS [43]. That is similar to our results when OCS15% with a bare appearance of OCS over 10 kDa in its poly-distribution, whereas OCS5% and OCS10% with a broader range of M_w_ and mean M_w_ closely to 10 kDa. The capability of OCS could be explained by a multi-facet explanation and much related to chitosan.

In terms of electrostatic interaction, with the deduction of M_w_, the -NH_2_ group, which carries a positive compartment of chitosan structure, can increase the ability to attach to the negatively charged bacterial cell wall, hence, enhancing the antibacterial characteristics. Moreover, the smaller the M_w_ of chitosan, the higher mobility of the polymer chains to bind to more colonies. Thus, the interaction between the bacteria and OCS happens and inactivates bacteria faster. This could be deduced that all the models of antibacterial of chitosan occurred, the bigger OCS fragments can attach to the cell wall then inhibit nutrient absorption. Likewise, the smaller fragments can diffuse through the cell membrane and bind to the DNA of bacterial, disturbing the mRNA transcription and protein synthesis. Furthermore, some researchers claimed that chitosan with a higher DD (over 90%) enhances stronger activity than lower DD (under 83.7%) [44]. The DD can contribute to the process due to the appearance of the more positively charged -NH_2_ groups in the chitosan structure [45].

The incorporation of OCS onto the EsPCL membrane could become a promising approach in the development of bioactive wound dressings. OCS15% was utilized to coat onto EsPCL due to its best water solubility. Moreover, since OCS15% has the lowest anti-microorganism effect in the agar diffusion test, it is reasonable to believe that other samples will perform better in terms of bacterial prevention. Due to the hydrophobicity of PCL, the adsorption could be minimized during the coating process. Thus, the membrane was hydrophilized using the plasma treatment that exposed polar functional groups of hydroxyl and carboxyl (-OH and -COOH) [46] on the surface of electrospun PCL fibers creating hydrogen bonds with water molecules. The adsorption efficiency of OCS on PCL fibers was evaluated with SEM. The SEM images show that the coating of OCS into EsPCL was successfully coated onto the membrane (Figure 3). The procedure of OCS adsorption onto the EsPCL membrane is similar to the procedure of Gel and Ag in our previous research [30]. In the first step, OCS has attached to PCL fibers thanks to the hydrogen interaction of -NH_2_ groups of OCS with both -OH and -COOH groups in the membrane surface. Then, when all the functional groups have interacted and the membrane is fulfilled by the OCS layer, the excessive OCS will be cross-linked with each other and increase the layer thickness. In addition, it is clear that the OCS formed into a layer in the membrane of which thickness increases correspondingly with the increase in coating times. When it comes to the agar disc diffusion results, the zones of inhibition show that C1 (mono-coating) has approximately no effects, and C6 (multi-coating) reveals the highest prevention against both types of bacteria. The difference between those membranes is mainly occurred by the amount of OCS released from the membrane. Therefore, the control of OCS coating on the EsPCL membrane is important and required further studies.

## 5. Conclusions

In this research, the chitosan and OCS derived from local Vietnamese sources were evaluated for their heavy metal contamination level, which was demonstrated to have adequate properties and high-yield purification for biomedicine applications. We employed H_2_O_2_ treatment under microwave irradiation to degrade chitosan into OCS with M_w_ from 2 to 7 kDa, in which the DD of chitosan was not affected due to the short irradiation time. The synthesized OCS were shown to possess not only strong inhibition against different skin-habitant microorganisms but also bactericidal effects against them. Furthermore, the use of tablet compression and coating methods for antibacterial testing is promising for evaluating the effects of antimicrobial agents in the powder formulation. The OCS15% sample possessed the most suitable molecular weight, water solubility, and antibacterial properties for further applications. Further investigation of incorporation of bioactive agents such as OCS and the EsPCL membrane should be considered regarding blood bleeding and actual wound healing effects on animal studies.

## Figures and Tables

**Figure 1 materials-14-04475-f001:**
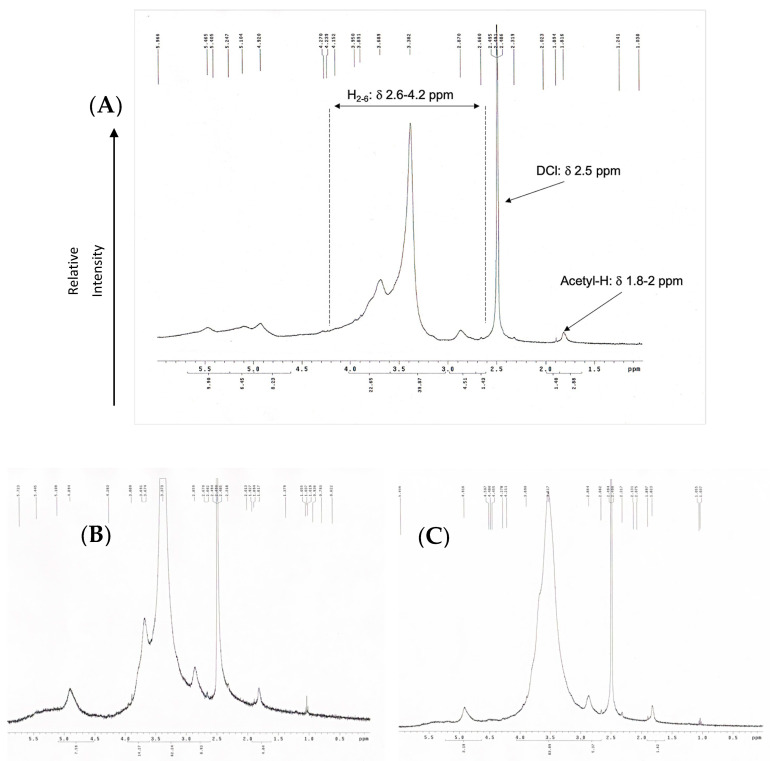
H-NMR (400 Hz) graphs of (**A**) OCS5%, (**B**) OCS10%, and (**C**) OCS15%.

**Figure 2 materials-14-04475-f002:**
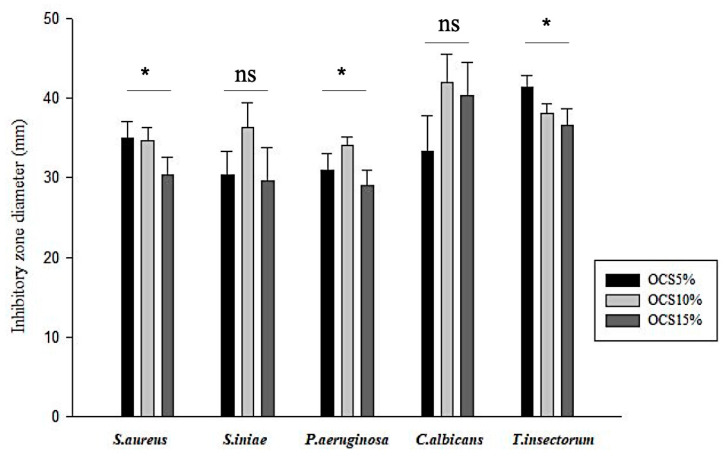
Measured inhibitory zone diameter of all samples (*n* = 3, (*) indicates *p* < 0.05, (ns) indicates *p* > 0.05.

**Figure 3 materials-14-04475-f003:**
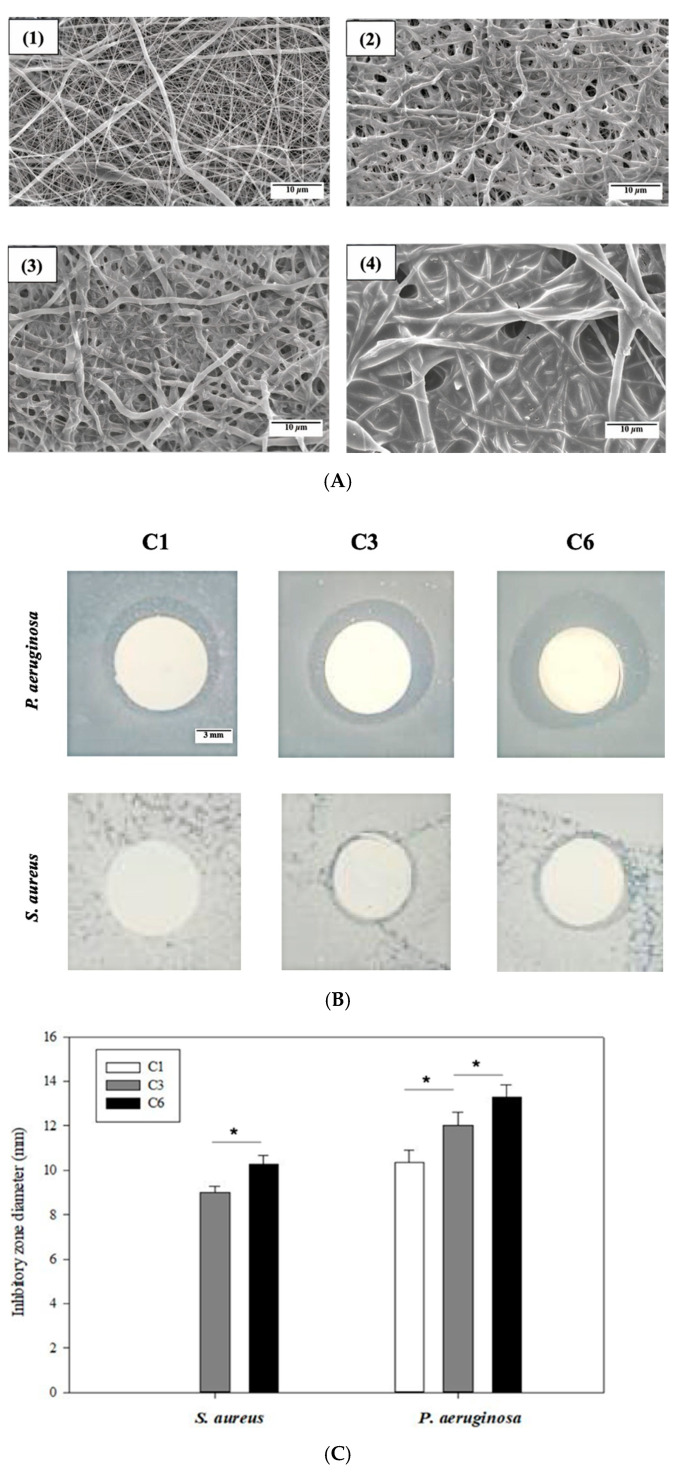
Images of (**A**) surface morphology of (**1**) EsPCL and EsPCLOCS membrane, (**2**) C1, (**3**) C3, and (**4**) C6, (**B**) inhibitory zone of EsPCLOCS membrane with different coating times, and (**C**) measured zone diameter against S. aureus and P. aeruginosa (*n* = 3, (*) indicates *p* < 0.05. The scale bar of Figure 3A is 10 μm, and Figure 3B is 3 mm.

**Table 1 materials-14-04475-t001:** Evaluation of heavy metals of chitosan and OCS by ICP-MS.

Sample	Pb (ppm)	As (ppm)	Hg (ppm)
Chitosan	0.052	0.05	ND *
OCS	ND *	ND *	ND *

ND: Not detected. * The minimum limit of detection is 0.02 ppm.

**Table 2 materials-14-04475-t002:** Weight average, number average, polydispersity index, and degree of deacetylation of OCS samples.

Sample	M_w_ (Da)	M_n_ (Da)	PDI	DD (%)	DE (%)
Chitosan	311,740	112,080	2.78	90.81	-
OCS5%	6878	2855	2.41	92.74	4532
OCS10%	4923	1924	2.5	92.44	6332
OCS15%	2474	1147	2.16	95.71	12,600

M_w_—weight average, M_n_—number average, PDI—polydispersity index, DD—degree of deacetylation, DE—depolymerization efficiency.

**Table 3 materials-14-04475-t003:** MIC and MBC results of OCS various types of microorganisms.

Microorganism	MBC (mg/mL)	MIC (mg/mL)
*S. aureus*	15	7.5 < [C] < 15
*S. iniae*	15	7.5 < [C] < 15
*P. aeruginosa*	7.5	3.75
*C. albicans*	15	7.5 < [C] < 15
*T. insectorum*	7.5	3.75 < [C] < 7.5

## Data Availability

The data used to support the findings of this study are included in the article.

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
