# Peer review of "Characterizations and Antibacterial Efficacy of Chitosan Oligomers Synthesized by Microwave-Assisted Hydrogen Peroxide Oxidative Depolymerization Method for Infectious Wound Applications"

_materials, 2021, doi:10.3390/ma14164475_

Round 1

Reviewer 1 Report

Very good piece of work. The application of Chitosan oligomers is a new aspect of the application of Chitosan as anti backterial agent. Accept in its present form. 

Author Response

We would like thank the reviewer for the comment and acceptance.

Reviewer 2 Report

The aim of research, which was shown in the manuscript entitled ‘Characterizations and antibacterial efficacy of chitosan oligomers synthesized by microwave-assisted hydrogen peroxide oxidative depolymerization method for infectious wound application’, was to determine an effect of H2O2 concentration applied on such properties of the obtained material as: depolymerization, molecular weight, degree of deacetylation, presence of heavy metals and the activity against some pathogens.

The obtained results are interesting and can be a basis for the further investigations. However, the manuscript needs to be improved before its publication.

Comments:

- Abstract: ‘Then, we examined the antibacterial inhibition’ - ‘Then, the antibacterial activity…was examined…

- Abstract: ‘while the molecular weight and degree of deacetylation of OCS were inversely proportional and proportional to the concentration of H2O2, respectively.’ There was shown in the manuscript that the deacetylation degree was not affected by the H2O2 concentration. This sentence should be corrected.

- Materials and methods: NMR: Was HCl or DCl used?

- Materials and methods: ICP-MS: Why were these metals selected for analysis? – Please, add this information.

-Results: What was the efficiency of OCS synthesis (per g of chitosan used) at particular concentrations of H2O2? Please, add this information.

-Results: Table 1: Were all OCS samples free from these metals? Please, add this information.

-Results: ‘Besides, Supplement S1 shows that the distribution of Mw of OCS5% and OCS10% is relatively the same with a considerably broad curve.’  - ‘relatively the same’ is not a proper description. Please, correct this. Moreover, by summing up the percentages of particular MW classes (for a given sample), a value much higher than 100% is obtained. Please, correct this (or the axis title in Fig. S1).

-Results: ‘The MIC of all the bacteria and fungi were not precisely determined’ – What was a reason for this?

-Results: ‘The EsPCL membrane before and after several OCS coatings were evaluated by SEM to confirm the absorption of OCS on EsPCL membrane’ – the adsorption of OCS on…

-Discussion: ‘These two oxidants are more powerful to cut the polymer chain through the following reactions [26]: ?2?2→?++ ???− ???−+ ?2?2→?2−∙+ ??∙ +?2?’. – These equations describe the process of formation of the active ingredients (oxidants). Please, correct this.

- The form: ‘we examined, we coated, etc.’ should be avoided

- Materials and methods: ‘Poly(ɛ-caprolactone) (PCL, Mn 80,000) were purchased’ -  Poly(ɛ-caprolactone) (PCL, Mn 80,000) was purchased…

Reviewer 3 Report

This article comprehensively describes characterization and antibacterial efficacy of chitosan oligomers synthesized by microwave-assisted hydrogen peroxide oxidative depolymerization method. The effects of hydrogen peroxide concentration on the physicochemical properties of chitosan oligomers were investigated through molecular weight, degree of deacetylation, and heavy metal contamination for optimization of chitosan oligomers formulation. Then, the authors examined the antibacterial inhibition and determined the minimum inhibitory concentration and minimum bactericidal concentration of chitosan oligomers based materials against common skin-inhabitant pathogens. The results show that chitosan based oligomer materials induce excellent antimicrobial properties (in terms of minimum inhibitory concentration and minimum bactericidal concentration). My first impression is that the research fits within the scope of the journal and the whole study can make an important contribution in the field of the chitosan oligomers research. The manuscript is submitted in grammatically and stylistically correct English. According to this, my suggestion would be that the manuscript may be published in the journal Materials in the present form.

Author Response

We would like to thank the reviewer for the comment and acceptance.

Reviewer 4 Report

The manuscript reports the preparation of oligomeric chitosans by depolymerization in H2O2 using microwave-irradiation. The antibacterial characteristics, including MIC and  MBC, and the morphology studies were performed  for the shortest oligomer (Mw ~ 2.5 Da) using 5 types of bacterial strains. For these studies the membranes based on electrospun PCL fibers were fabricated and then coated with oligomer giving from mono- to multi-layers. The manuscript should be of interest to scientists, but before publication it needs minor revision.
My comments concerning this work are following: 
Brief description of deacetylation as the kind of chitosan modification should be added in Section 2.2.1 to introduce the regulation way of deacetylation degree (as it is made for preparation of membranes in section 2.2.3. Please, explain why shorter chitosan oligomers were not tested.
Authors highlighted that the chitosan becoming from local Vietnamese shrimps living in Mekong Delta. What is the difference between chitosan obtained from other sources than the one used for research?
It is stated that the studied membranes are a potential wound dressing materials. In what form will they be used? sticky patches, hydrogel layers or others?
DD abbreviation appears on p. 4, whereas it is explained on p. 8.
The brand of ICP-MS is missing in Experimental part.
